# Genomic Diversity of Common Sequence Types of *Listeria monocytogenes* Isolated from Ready-to-Eat Products of Animal Origin in South Africa

**DOI:** 10.3390/genes10121007

**Published:** 2019-12-04

**Authors:** Itumeleng Matle, Rian Pierneef, Khanyisile R. Mbatha, Kudakwashe Magwedere, Evelyn Madoroba

**Affiliations:** 1Bacteriology Division, Agricultural Research Council: Onderstepoort Veterinary Research, Onderstepoort, Pretoria 0110, South Africa; 2Department of Agriculture and Animal Health, Science Campus, University of South Africa, Private Bag X6 Florida 1709, South Africa; mbathkr@unisa.ac.za; 3Biotechnology Platform, Agricultural Research Council, Onderstepoort Veterinary Research, Onderstepoort, Pretoria 0110, South Africa; PierneefR@arc.agric.za; 4Directorate of Veterinary Public Health, Department of Agriculture, Forestry and Fisheries, Private Bag X 138, Pretoria 0001, South Africa; KudakwasheM@daff.gov.za; 5Department of Biochemistry and Microbiology, Faculty of Science and Agriculture, University of Zululand, Private Bag X1001, KwaDlangezwa 3886, South Africa; evelyn.madoroba@gmail.com

**Keywords:** ready to eat, meat products, prophage, *Listeria monocytogenes*, virulence, resistance, *Listeria* pathogenicity islands and Stress Survival Islet diversity

## Abstract

*Listeria monocytogenes* is a highly fatal foodborne causative agent that has been implicated in numerous outbreaks and related deaths of listeriosis in the world. In this study, six *L. monocytogenes* isolated from ready-to-eat (RTE) meat products were analysed using Whole Genome Sequencing (WGS) to identify virulence and resistance genes, prophage sequences, PCR-serogroups, and sequence types (STs). The WGS identified four different STs (ST1, ST121, ST204, and ST876) that belonged to serogroup 4b (lineage I) and 1/2a (lineage II). Core genome, and average nucleotide identity (ANI) phylogenetic analyses showed that the majority of strains from serogroup 4b (lineage I) clustered together. However, two isolates that belong to serogroup 1/2a (lineage II) grouped far from each other and the other strains. Examination of reference-guided scaffolds for the presence of prophages using the PHAge Search Tool Enhanced Release (PHASTER) software identified 24 diverse prophages, which were either intact or incomplete/questionable. The National Center for Biotechnology Information- Nucleotide Basic Local Alignment Search Tool (NCBI-BLASTn) revealed that *Listeria monocytogenes* strains in this study shared some known major virulence genes that are encoded in *Listeria* pathogenicity islands 1 and 3. In general, the resistance profiles for all the isolates were similar and encoded for multidrug, heavy metal, antibiotic, and sanitizer resistance genes. All the isolates in this study possessed genes that code for resistance to common food processing antiseptics such as Benzalkonium chloride.

## 1. Introduction

*Listeria* species are ubiquitous bacteria widely distributed in the environment of which *Listeria monocytogenes* is the most important zoonotic species of global public health and economic importance in the genus [1]. The general approach to prevent listeriosis in the human population is to restrict the exposure of the human and animal populations to foods contaminated with *L. monocytogenes*; however, in the case of a listeriosis outbreak, timeous removal of suspected foods during ongoing epidemiological investigations limits human exposure and spread of the disease [2,3]. Identification of suspected contaminated food during a listeriosis outbreak is primarily performed by patient interview, a process that is long and is usually hampered by several factors such as low incidence of *L. monocytogenes*, the ubiquitous presence of *L. monocytogenes* in the environment, and the wide variation of the incubation time which usually range from 3–90 days [4,5]. These drawbacks are exacerbated by limitations in patient memory during interviews and, in some instances, the inability to conduct effective interviews [6]. 

Recently, molecular-based subtyping comparisons to match human isolates to food or environmental isolates have become critical for tracking and source identification of the cause of outbreak [7]. Traditionally, pulsed-field gel electrophoresis (PFGE) has been used as the “gold standard” for subtyping of *L. monocytogenes* isolates involved in outbreaks and sporadic cases; however, Whole Genome Sequencing (WGS) has emerged as a powerful tool for subtyping and investigation of *L. monocytogenes* outbreak cases [8]. Typing in WGS is performed at higher resolution than that of traditional molecular typing methods as it uses the entire genome of a bacterium and, consequently, WGS can reveal the genetic differences between the sequence types, the acquisition, and evolution of virulence as well as the pathogenic traits and antimicrobial resistance profiles of *L. monocytogenes* [9]. In 2013, the United States employed WGS as a primary method for subtyping of *L. monocytogenes*, which led to the identification of more outbreaks than could have not been detected by PFGE. The use of WGS further led to the differentiation between strains with indistinguishable PFGE profiles which enhanced resolution in the outbreak investigations [10]. Moreover, WGS is important in helping to understand the biology, phylogeny, and ecology of *L. monocytogenes* contamination in the food value chain [11]. 

The cost reduction of WGS has allowed it to become the preferred method for molecular subtyping of *L. monocytogenes* outbreaks and a viable alternative tool for the source attribution of listeriosis cases [12,13]. Apart from two studies that reported on the use of WGS for typing of *L. monocytogenes*, which was associated with human listeriosis outbreaks in 2015 and 2018 [14,15], there is no published genomic information on this pathogen in South Africa. Considering that listeriosis is a notifiable human disease in many countries including South Africa and considering its association with ready-to-eat (RTE) food products, a need exists to generate more genomic data on *L. monocytogenes* obtained from RTE products for epidemiological purposes such as source identification and tracking. Polony and biltong are the most popular RTE meat products in South Africa, accounting for approximately up to 50% of the country RTE meat product production [16,17]. Therefore, the aim of this study was to characterise the strains of *L. monocytogenes* isolated from RTE meat products in South Africa. The WGS information of the *L. monocytogenes* strains was analysed in order to identify virulence and resistance genes, prophage sequences, phylogeny, PCR-serogroup, and sequence type (ST). 

## 2. Materials and Methods 

### 2.1. Sample Information 

The samples used in this study were collected from supermarkets and butcheries located in four provinces of South Africa, namely Gauteng, Limpopo, Mpumalanga, and Western Cape, as indicated in Figure 1 as part of the routine national survey for *L. monocytogenes* in meat and meat products in South Africa [18]. Isolates of *L. monocytogenes* from biltong (n = 5) and Polony (n = 1) samples were sequenced in this study. Samples were collected aseptically between 2015 and 2016 using sterile plastic bags and transported on ice immediately to the Onderstepoort Veterinary Research (OVR): Feed and Food laboratory, SA for microbiological analysis. 

### 2.2. Microbiological Analysis 

Microbiological analysis of the samples was performed according to procedure described by Matle et al. [18]. Briefly, samples weighing 25 g each were aseptically transferred into 225 mL of ONE broth-*Listeria* (Oxoid, Basingstoke, UK), followed by homogenization for 2 min using a Stomacher (Stomacher Lab Blender 400, Seward Ltd., West Sussex, UK). After homogenization, the broth sample was incubation at 35 °C for 24 hours. The broth samples (10 µL per sample) were inoculated onto Brilliance-*Listeria* plates (Oxoid, Basingstoke, UK) and incubated at 35 °C for 24 hours. Presumptive *Listeria* colonies were subjected to Oxoid Biochemical Identification System (Oxoid, Basingstoke, UK) for identification. The isolates that were confirmed as *L. monocytogenes* were preserved in brain–heart infusion (Oxoid, Basingstoke, UK) broth supplemented with 35% glycerol and stored at −80 °C at OVR: Feed and Food laboratory.

### 2.3. Genomic Deoxyribonucleic Acid (DNA) Extraction

DNA was extracted using the High Pure Polymerase Chain Reaction (PCR) Template preparation kit (Roche, Potsdam, Germany) as per manufacturer’s protocol. Briefly, pure colonies of *L. monocytogenes* on blood agar (Oxoid, Basingstoke, UK) were inoculated into 50 µL of DNA-free water followed by adding 200 µL of binding buffer and 40 µL of Proteinase K. The mixture was then incubated at 70 °C using heating block for 10 min. After incubation, 100 µL of isopropanol was added and the mixture was applied to a High Pure Filter tube followed by centrifugation at 13,000 rpm for a minute. The flow-through and collection tube were discarded. Then, 500 µL of inhibitor removal buffer was added on the High Pure Filter tube followed by centrifuging at 13,000 rpm for a minute and discarding of the flow-through and collection tube. The filter tube was washed two times with 500 µL of wash buffer and centrifuged at 13,000 rpm for a minute. Then, 200 µL of elution buffer (70 °C) and new collection tubes were added, followed by centrifuging at 13,000 rpm for a minute. The extracted DNA was stored at −80 °C for further analyses. The DNA quantity and purity were assessed by using Qubit fluorimetric quantitation (Thermo Fisher Scientific, Waltham, MA, USA). 

### 2.4. Genome Sequencing and De Novo Assembly

Whole genome sequencing of the samples was performed at the Biotechnology Platform, Agricultural Research Council, Onderstepoort, South Africa. The DNA libraries were prepared using the Nextera XT DNA library preparation kit (Illumina, San Diego, CA, USA), followed by 2 × 300 paired-end sequencing on a MiSeq instrument (Illumina, San Diego, CA, USA). Quality control, including adapter removal of the raw data, was done using Trimmomatic [19]. SPAdes “careful” mode was used to create a de novo assembly of each isolate [20]. All de novo assembled contigs were compared to the reference *L. monocytogenes* EGD-e chromosome, complete genome (NC_003210.1) using BLAST Ring Image Generator (BRIG) [21].

### 2.5. Core Genome Determination

Gene prediction in protein format was done for all contigs using PROkaryotic DYnamic programming Gene-finding Algorithm (Prodigal) [22], with the “closed ends” parameter specified to ensure the prediction of complete proteins. A total of 37 other *Listeria* complete genomes (Table 1) were obtained from the NCBI RefSeq database [23] to aid in the construction of a representative core genome. The PCR-serogroup, lineage, and Multilocus sequence typing (MLST) information for these strains were obtained from the *Listeria* database hosted by the Pasteur Institute, France (http://bigsdb.pasteur.fr/listeria/) [24]. All proteins from the 37 complete genomes and those predicted in our strains were compared by means of an all-against-all Protein BLAST (BLASTp) [25] with an e-value cut-off set at 1 × 10^−30^. The Markov Clustering (MCL) algorithm was used to cluster proteins in nonoverlapping groups [26]. For the MCL algorithm, the pairwise BLASTp bit-scores were used as edge weights and the inflation parameter was set at 1.8. Clusters containing single-copy orthologs from all strains as well as strains in this study were used for further phylogenetic analysis.

### 2.6. Core Genome Phylogenetic Analysis

Core genome clusters containing single-copy orthologs from all strains, including the controls, were used for core genome phylogenetic analysis. For each cluster, the protein sequences from all genomes were aligned using Multiple Alignment using Fast Fourier Transform (MAFFT) [27] with the default parameter “auto”. All multiple sequence alignments were concatenated into a single file, which was used for phylogenetic inference by the program IQ-TREE [28] with the “ModelFinder Plus” parameter to calculate the best-fit substitution model, and 1000 bootstrap replicates were specified. The HIVb+F+R2 was chosen as the best-fit model according to BIC. The resulting consensus tree was visualized and edited using the Interactive Tree of Life (iTOL) [29].

### 2.7. Prophage Identification and Analysis

Reference-guided scaffolding was done by means of an algorithm for genome scaffolding called Multi-Draft based Scaffolder (*MeDuSa*) [30] using *L. monocytogenes* EGD-e chromosome, complete genome (NC_003210.1) as a reference. The largest scaffold for each strain was used to predict putative prophages with PHAge Search Tool—Enhanced Release (PHASTER) [31].

### 2.8. Determination of Plasmids 

In order to determine the plasmids, the contig files of the de novo assembled genomes from this study were analysed with Plasmid Finder 2.0 for the specified Gram-positive scheme [32]. 

### 2.9. Average Nucleotide Identity Calculation

Average nucleotide identity (ANI) was determined using ANI calculator, using both best hits (one-way ANI) and reciprocal best hits (two-way ANI) between genomic datasets as described [33]. In order to obtain normalized ANI values, a mean of pairwise ANI values were calculated from those obtained for all pairs between strains taking cognisance that the ANI values between genomes of the same species must be above 95%. Statistics on the distribution of ANI values were analysed using the R statistical software (http://www.r-project.org).

### 2.10. In Silico PCR-Serogroup and ST Prediction

PCR-serogroup and Multi Locus Sequence Type (MLST) profiles were obtained from the *Listeria* database hosted by the Pasteur Institute, France (http://bigsdb.pasteur.fr/listeria/) [24]. The PCR-serogroup database contains 5 loci with 142 different alleles, and the MLST database contains 7 loci with a total of 1799 different alleles. The presence and combination of these different alleles allow for the prediction of *Listeria* PCR-serogroups and MLST. A k-mer based, mapping tool, stringMLST [34] was used to align reads against these profiles to determine the PCR-serogroup and MLST for each sample using k-mers of length 35. To validate the k-mer-based predictions, all contigs were further compared against the PCR-serogroup and MLST profiles by BLASTn [25] with an e-value cut-off set to 1 × 10^−30^ and with allele presence and combination determined.

### 2.11. Virulence Factors

All predicted proteins for the samples were compared with the Virulence Factor Database (VFDB) [35] by means of BLASTp with an e-value cut-off set at 1 × 10^−5^ and a minimum identity of 75%. Some of the main virulence factors were examined using the well-annotated *L. monocytogenes* EDGe as a reference genome. Virulence factor intersections between samples were visualized with the R package UpSetR [36].

### 2.12. Resistance Profiles

Antibiotic and antibacterial biocide and metal-resistance profiles for the samples from this study were determined by comparing all the predicted proteins to the BacMet [37], MEGARes [38] and nonredundant antibiotic resistance database (noradab) (noradab.bi.up.ac.za) databases, respectively. This was done by means of BLASTp and tBLASTn for the MEGARes database which is in nucleotide format with an e-value cut-off set at 1 × 10^−5^ and a minimum identity of 75%.

### 2.13. Data Availability

The six genome sequences of *L. monocytogenes* isolates were deposited at the National Centre for Biotechnology Information (NCBI)/GenBank under the accession numbers from SAMN12360665 to SAMN12360670 (BioProject No. PRJNA556582).

## 3. Results

### 3.1. Genome Sequencing, Assembly, and Annotation

All de novo assembled contigs from genomes from this study were aligned and compared with *L. monocytogenes* EGD-e chromosome, complete genome (NC_003210.1) as a reference by means of BRIG (Figure 2). Visualization with BRIG indicates that the genomes displayed typical attributes of *L. monocytogenes* such as assembly sizes ranging from 2.9 to 3.1 base pairs and low genomic G+C content of 37.8%. The difference in assembly sizes was due to variation in the length of prophages from the sequenced isolates (Figure 3). De novo assembly ranged from 38 to 47 contigs with N50 between 411,134 bp and 637,980 bp (Table 2). 

### 3.2. Plasmid Identification 

Plasmid identification was performed to determine if the sequenced isolates in this study harboured plasmids that confer resistance of *L. monocytogenes* to antibiotics. The analysis of the plasmids revealed that none of the isolates possessed known plasmids associated with *L. monocytogenes* (Table 2).

### 3.3. Stress Survival Islet 

The stress survival islets (SSI-1 and SSI2) are known to be responsible for the proliferation of *L. monocytogenes* under stressful conditions in food processing facilities [39,40]. In the current study, SSI-1 was detected in a E313 sample belonging to ST1 of lineage I. SSI-2 was present in two samples (E367 and E362) which belong to ST121 and ST204 of lineage II, respectively (Table 2).

### 3.4. Listeria Pathogenicity Islands

In this study, each of the 6 sequenced samples encoded Listeria pathogenicity islands 1 (LIPI1), which houses a cluster of six virulence genes of *L. monocytogenes* (Table 2). LIPI3, which is a gene cluster that encodes a potential haemolytic factor (Streptolysin S), was present in samples E258, E313, E359, and E916. Samples E258, E313, E359, and E916 belonged to serogroup 4b and lineage I. The LIPI3 was absent in samples E362 and E367 of serogroup 1/2a and lineage II. LIPI-4, which has been recently described as a gene cluster involved in neural and placental infection, was not detected in this study.

### 3.5. Multi-Locus Sequence Typing (MLST) and PCR-Serogroups

The MLST analysis, which is used to identify sequence types, lineages, and clone complexes of closely related *L. monocytogenes* strains, identified four different STs (Table 3). ST1 represented 50% (n = 3/6) of the samples while the other 50% was represented by ST121, ST204, and ST876. When silico PCR-serogroup analysis was used to differentiate among major serotypes of *L. monocytogenes*, it revealed that the strains from this study belonged to 4b (lineage I) and 1/2a (lineage II).

### 3.6. Core-Genome Phylogenetic Analysis

A total of 124,729 proteins were used for the generation of orthologous clusters which resulted in a core genome size of 1753. Core genome phylogenetic analysis reveals that the majority of samples (E252, E313, E359, and E916) in this study belonged to serogroup 4b (lineage I) clustered together (Figure 4). Two samples (E362 and E367) that were assigned to serogroup 1/2a (lineage II) grouped far from each other compared to the rest of the samples. The core genome phylogenetic analysis was in correlation with the predicted PCR-serogroups and MLST.

### 3.7. Average Nucleotide Identities

Average Nucleotide Identity (ANI) is a measure of nucleotide-level genomic similarity between the coding regions of two genomes. The ANI phylogenetic analysis corroborated the core genome phylogeny analysis and MLST prediction results. The analysis indicated that samples E258 (1544), E313 (1545), E359 (1545), and E916 (1549) clustered together, while samples E362 (1547) and E367 (1548) were far from all the other samples (Figure 5). Sample E359 was sequence typed as ST876 from lineage I, which belong to serogroup IVb on MLST; however, on ANI (Figure 5), it clustered with most of the ST1s that belongs to the same lineage and serogroup.

### 3.8. Virulence Factors 

The distribution of main virulence genes of *L. monocytogenes* was surveyed in this study. A total of 142 virulence genes were identified across all six sequenced *L. monocytogenes* samples. Each sample contained between 80 and 86 virulence genes. Forty-six similar virulence genes were present in all the six sequenced strains. The NCBI-BLASTn revealed that samples from this study share some of the known major virulence genes. Those similar virulence genes include *prfA, plcA, plcB, hly, iap/cwha, iapB, actA*, and *mpl* as well as the internalin AB operon, which is encoded in LIPI1. However, other internalin family member genes (*inlF, InlC, inlK*, and *InlJ*) as well the genes *vip, ami, gtcA, igt, oat, pdgA*, *agrA, agrC, prsA2, oppA, hbp2, srtA,* and *srtB* were identified in all the samples. 

Interestingly, samples E362 and E367 belonging to lineage II and serogroup 1/2a harboured the largest number of unique virulence genes and contained 21 similar genes, which were not present in the other samples. The unique virulence genes were shared by samples including the *inlGHE* cluster among others. The virulence genes associated with LIPI3 were not identified in samples E362 and E367. The LIPI3-associated genes were mostly identified in samples belonging to lineage I and included haemolysin known as listeriolysin S (LLS)*, llsB, llsG, llsY, llsX*, and *llsD* genes. The truncated internalin A (*inlA*) gene and deletion in the ActA protein were not observed in all the samples. 

### 3.9. Prophage Identification and Analysis

Reference-guided scaffolding resulted in near full-length assemblies for each sample, which allowed for the prediction of putative prophages (Table 4). A total of 24 prophages (either intact or incomplete/questionable) were identified in all genomes by PHASTER software including one intact prophage identified in sample E258 and two intact prophages identified in samples E313 and E362 (Table 4). Samples E258 and E916 harboured three intact prophages each, while no intact prophage was identified in E367. Furthermore, incomplete and questionable prophages were identified among various samples in this study. Sample E362 carried the highest number of incomplete prophages, followed by E258, E313, and E367 with single incomplete prophage each. Questionable prophages were observed in all the samples except E362. The Phage Classification Tool Set (PHACTS) [41] also gives a possible phage source for the intact prophages with PHAGE_Lister_ vB_LmoS_188 and PHAGE_Lister_vB_LmoS_293 phages being predicted in the majority of the samples. Other intact prophages that were identified included PHAGE_Lister LP030-2, PHAGE_Lister B054, PHAGE_Lister A006, and PHAGE_Lister LP101.

### 3.10. Resistance Genes

In general, the resistance profiles for all the samples sequenced in this study were similar. The examination of heavy metal, multidrug, and antibiotic resistance genes revealed the presence of resistance genes that coded for multidrug resistance genes (*EmrB/QacA* and *Bcr/CflA* family) quaternary ammonium compound resistance (*SugE, Tn6188,* and *bcrABC*), fosfomycin resistance (*fosX*), lead/cadmium/zinc resistance, copper resistance (*CopC*), cobalt/zinc/cadmium resistance, and aluminium resistance (*CorA* and *CzcD*) in all samples. Other resistance genes included *tetA, tetM, mecC, mrB, msrA, lde*, and *mdrL*.

## 4. Discussion

The current study characterised the genomic diversity of six *L. monocytogenes* isolates that were isolated from biltong (E258, E313, E359, E362, and E367) and polony (E916) samples collected in four provinces of South Africa between 2015 and 2016. The alignment of samples from this study to *L. monocytogenes* EGD-e reference strain displayed features that are in line with those found in most *Listeria* genomes as reported in previous studies by Den Bakker et al. [4] in the US, by Kuenne et al. [42] in Germany, and by Schmitz-Esser et al. [43] in Austria. These features include assembly size, N50, and genomic G+C content. However, the difference in assembly size (2.9 to 3.1 base pairs) among the genomes might be influenced by the presence of prophages as their length varied and did not occur in the same places in the genomes in this study.

The application of MLST to subtype *L. monocytogenes* has provided important information into the population structure of this pathogen. *Listeria monocytogenes* sequence types reported in this study have been shown to have global distribution [44,45,46]. *Listeria monocytogenes* ST1 comprised 50% of all samples in this study and is known to be overrepresented in clinical and food isolates in the world as previously reported [43]. *Listeria monocytogenes* ST121 and ST204 are regarded as the most common persistent strains in food-processing environments [47]. These sequence types have the ability to survive and persist for months and even years in food-processing environments and to keep contaminating food products [46]. Importantly, the STs reported in the current study were also identified in human listeriosis cases associated with the 2017–2018 outbreak in SA [15]. 

Phylogenetic analysis was used to determine the genetic relationship among the samples from different sources, geographic areas, and time by inferring evolutionary relatedness [46]. Two different phylogenetic analyses were employed to establish relatedness amongst strains in the present study to other international strains. The core genome and ANI phylogenetic analyses revealed similar results, indicating that majority of the samples (E258, E313, E359, and E916) belonged to serogroup 4b, lineage I, and ST1 except E359, which belonged to ST876 clustered together. Based on ANI results, ST876 and ST1 shared an ANI of >95% similarities and all belonged to lineage I. This suggests that the *L. monocytogenes* isolates might have a common ancestor, hence the clustering. Further phylogenetic analyses indicated that two of the samples (E362 and E367) belonging to serogroup 1/2a and lineage II grouped far from each other and the rest of the strains. The MLST validated this clustering as samples E362 and E367 belong to ST121 and ST204, respectively.

The genomes from the present study were compared with 37 global genomes from different countries. Importantly, it was observed that samples E258, E313, E359, and E916 clustered with *L. monocytogenes* strains isolated from fatal outbreaks, clinical cases, food, and environment. Furthermore, samples E362 and E367 clustered with known important international strains of *L. monocytogenes* such as *L. monocytogenes* EGD-e, *L. monocytogenes* 36_25-1, and *L. monocytogenes*_F6654. This observation suggests that a theoretical virulence potential exists and that the samples could cause diseases in humans in South Africa. 

*Listeria monocytogenes* is known to carry several plasmids that often confer resistance to antibiotics and to increase stress tolerance [42]. Those plasmids include pLM4423, pLM6179, and pIP501, which code for common features found in *Listeria* spp such as the CadAC Cadmium resistance transposon Tn5422 [42]. The analysis of the plasmids identified that none of the strains in this study carried a known plasmid. Plasmids are movable elements, and bacteria can gain or lose them; therefore, their absence in our strains is not surprising. Furthermore, Reference [48] reported *L. monocytogenes* isolates that possess no plasmids in their study. 

Examination of reference-guided scaffolds for the presence of prophages using the PHASTER software identified 24 diverse prophages. However, the distribution of these prophages was uneven among the samples, as some samples contained intact prophages while others had questionable or incomplete prophages. The PHASTER also identified about 11 intact prophages from 6 different *Listeria* phages (LmoS188, LmoS293, A006, LP101, LP303-2, and B054) and gave the possible source for the phages in which the prophage was detected. The intact prophages detected in this study either belong to *siphoviridae* (A006, A118, B054, LP 100, and LP 030-3) and *myoviridae* (LmoS188 and LmoS293) *Listeria* phages as previously reported in other studies [42,49,50,51]. The identified phages in the present study are commonly found in *Listeria* strains globally and have been associated with the survival evolution and persistence of *L. monocytogenes* in food-processing facilities [42,52]. The virulence and pathogenicity of *L. monocytogenes* is also influenced by the presence of these prophages [53].

The WGS can also be used to identify the presence of pathogenicity islands linked to virulence or particular modes of pathogenesis [47]. In the present study, each of the six *L. monocytogenes* samples encode LIPI-1, which harbour *Prf-A* dependent virulence cluster genes that are critical in the infectious cycle of *L. monocytogenes* [1]. This could explain the reason why all sequenced isolates share about 40% similar genes that are strongly associated with LIPI-1. The genes encoded by LIPI-1 a common feature often found in RTE food and food production environment: *L. monocytogenes* isolates [54]. The LIPI-3 was identified only in strains that belong to lineage I and serogroup 4b, and it encodes a haemolytic factor (lysteriolysin, LLS), which increases the virulence of *L. monocytogenes*. LIPI3 is overrepresented among lineage I strains in human isolates and is strongly associated with serotype 4b isolates [55]. The absence of LIPI3 in two samples of lineage II was expected as the sequence types have been reported to lack it LIPI [1]. The LIPI4 is a newly described *Listeria* pathogenicity island [56] which encodes for the genes that annotated as a cellobiose family PTS system [55], and it was not identified in any of the strains identified in this study. The LIPI4 appears to be strongly associated with ST4 isolates [1]; hence, it was not observed in the current study. Several studies have identified mutations that are important in *L. monocytogenes* virulence markers [40,56,57,58]. The analysis of all sequenced strains in the present study indicated that they all encode for full-length InlA and ActA proteins.

Examination of resistance genes revealed that all sequenced strains of *L. monocytogenes* in the current study contained identical genes. The tetracycline resistance genes (*tetA, tetL, tetT*, and *tetM*) were found in all the samples, which have been identified in strains isolated from various meat samples [57]. Moreover, all the *L. monocytogenes* isolates in this study carried the *mecC* gene that is known to confer resistance to beta-lactam drugs, which is the primary therapeutic option for human listeriosis together with aminoglycosides [16]. The genes encoding for lincomycin resistance protein (*lmrB*), fosfomycin resistance protein (*fosX*), and erythromycin resistance ATP-binding protein (*msrA*) were also identified across all samples. The efflux pump-related genes, *lde* and *mdrL*, which confer resistance to quinolone and macrolides, respectively, were also identified. Different multidrug resistance transporter and efflux pump proteins that are known to confer resistance in *L. monocytogenes* and other bacteria [59] were observed in this study.

Quaternary ammonium compounds (QACs) such as benzalkonium chloride (BC) are widely used for cleaning and disinfection of food-processing environments [60]. The known molecular mechanisms of BC resistance are due to the activity of efflux pump systems encoded through the brcABC [61] and qacH on the Tn6188 transposon [62] genes that can be acquired through horizontal gene transfer leading to BC resistance in *L. monocytogenes*. In this study, QAC resistance genes (*SugE, Tn6188*, and *BcrA*) were identified, which suggests that these isolates are adapted to survival in the food-processing environment where sanitizers such as QACs are commonly used [62]. The BC is recommended in many countries as the most effective disinfectant against *L. monocytogenes* in food processing facilities [63]; however, the presence of *BcrA* and *Tn*6188 genes in strains from this study poses a serious hygiene management concern. Therefore, a need exists to evaluate the use of BC sanitary products in food-processing facilities in South Africa.

*Listeria monocytogenes* is equipped with mechanisms that allow it to adapt to and survive under stressful conditions [39]. However, there is a high degree of strain divergence in stress response and environmental adaptation, which is mostly associated with the presence of SSI-1 and SSI-2 [40]. According to Reference [39], SSI-1 supported the growth of *L. monocytogenes* under salt, acidic, bile, and gastric stress conditions and is overrepresented in human clinical isolates, whereas SSI-2 is mostly found in isolates from food and food-processing environments. In the current study, SSI-1 was found in the isolates while SSI-2 was detected from only two isolates that belonged to the sequence types ST121 and ST204, which are known to be persistent in food-processing environments [64]. The presence of SSI-2 in ST121 and ST204 can be suggested to be the factor for the persistence of these strains [63].

## 5. Conclusions

*Listeria monocytogenes* is one of the most well-characterized foodborne pathogens in the world; however, in South Africa, there is lack of information on the molecular characterisation of the pathogen in meat and RTE products. The current study is the first to report on the characterisation of *L. monocytogenes* strains isolated from biltong and polony products using whole genome sequencing in South Africa. The application of WGS in the current study has provided a partial overview of genomic diversity of *L. monocytogenes* strains that are circulating in South African RTE meat products. It also adds to the global data on genetic diversity of *L. monocytogenes* from South Africa. This overview has revealed virulence potential associated with presence of cluster LIPI-1 genes in all the isolates. In addition, it has permitted the evaluation of resistance genomic characteristics of the isolates, which indicate the presence of resistance genes to common antimicrobial agents and sanitizer. Prophages, which were observed in South African isolates, are largely found across the global clonal diversity of *L. monocytogenes*, most probably due to the influx of imported raw materials such as mechanically separated meat (MSM), which commonly is used in the manufacturing of polony. The information provided in this study is important for enhancing the understanding of evolutionary divergence, adaptation, and survival of *L. monocytogenes* in products of animal origin.

## Figures and Tables

**Figure 1 genes-10-01007-f001:**
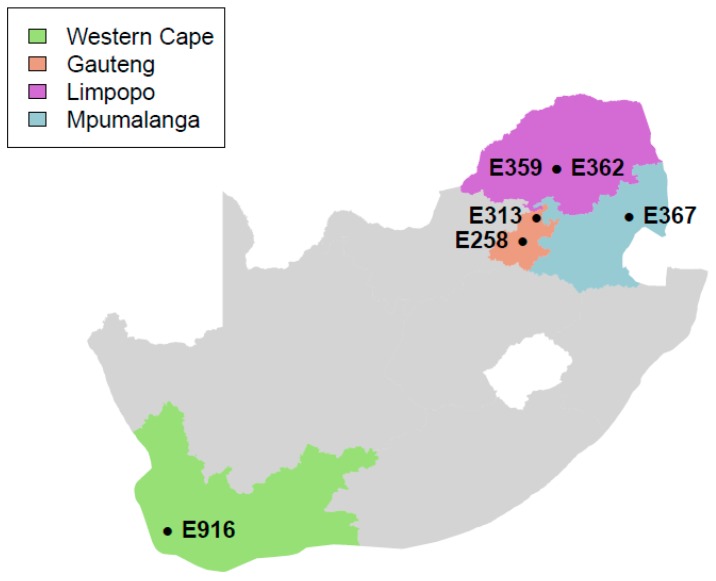
Location of supermarkets and butcheries from which samples were collected in South Africa.

**Figure 2 genes-10-01007-f002:**
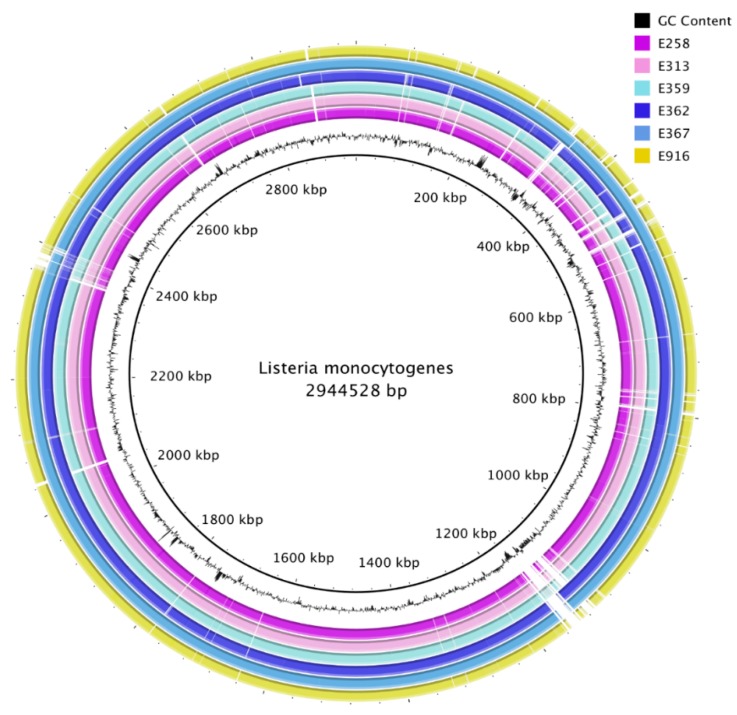
Comparison of ready-to-eat (RTE) sample contigs against the *Listeria monocytogenes* EGD-e chromosome, complete genome (NC_003210.1), displayed in the inner black circle.

**Figure 3 genes-10-01007-f003:**
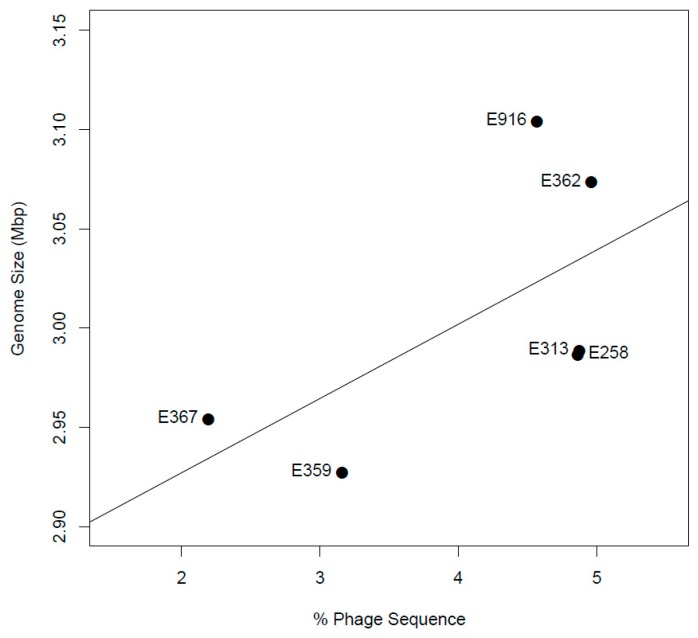
The correlation of genome size with percentage of phage sequence.

**Figure 4 genes-10-01007-f004:**
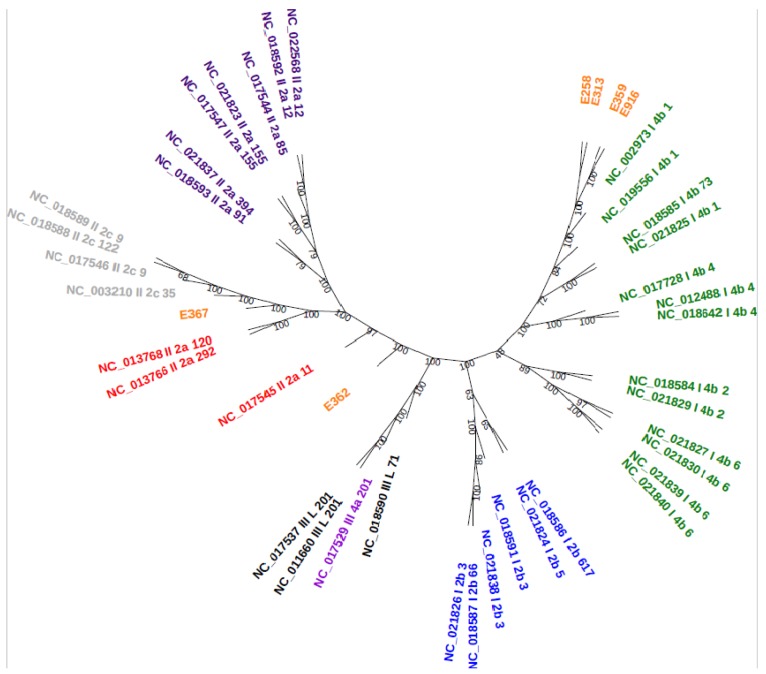
Core genome phylogenetic tree: RefSeq accessions are followed by lineage, PCR-serogroup, and MLST. The RTE samples are indicated in orange.

**Figure 5 genes-10-01007-f005:**
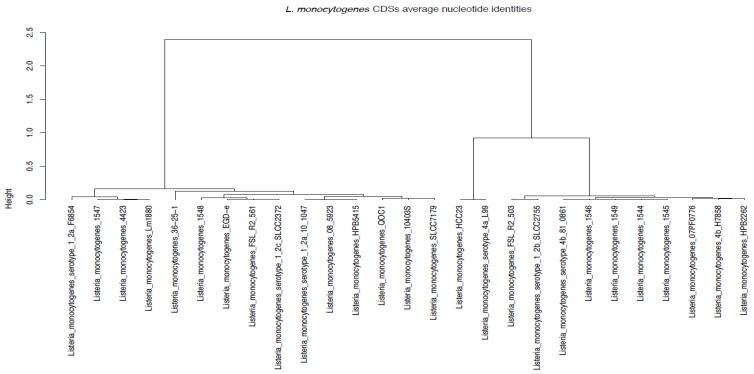
*Listeria monocytogenes* CDS average nucleotide identities using Gower distance metric.

**Table 1 genes-10-01007-t001:** List of *Listeria monocytogenes* strains used for the construction of a core genome.

Strain	* MLST-ST	Lineage	Serogroup	* NCBI RefSeq Accession Number
**EGD-e**	35	II	2c	NC_003210
**07PF0776**	4	I	4b	NC_017728
**08-5578**	292	II	2a	NC_013766
**08-5923**	120	II	2a	NC_013768
**10403S**	85	II	2a	NC_017544
**ATCC 19117**	2	I	4b	NC_018584
**C1-387**	155	II	2a	NC_021823
**Clip81459**	4	I	4b	NC_012488
**F2365**	1	I	4b	NC_002973
**FInlAnd 1998**	155	II	2a	NC_017547
**FSL R2-561**	9	II	2c	NC_017546
**HCC23**	201	III	L	NC_011660
**J0161**	11	II	2a	NC_017545
**J1-220**	6	I	4b	NC_021830
**J1776**	6	I	4b	NC_021839
**J1816**	2	I	4b	NC_021829
**J1817**	6	I	4b	NC_021827
**J1926**	6	I	4b	NC_021840
**J2-031**	394	II	2a	NC_021837
**J2-064**	5	I	2b	NC_021824
**J2-1091**	1	I	4b	NC_021825
**L312**	4	I	4b	NC_018642
**L99**	201	III	4a	NC_017529
**LL195**	1	I	4b	NC_019556
**M7**	201	III	L	NC_017537
**N1-011A**	3	I	2b	NC_021826
**R2-502**	3	I	2b	NC_021838
**SLCC2372**	122	II	2c	NC_018588
**SLCC2376**	71	III	L	NC_018590
**SLCC2378**	73	I	4b	NC_018585
**SLCC2479**	9	II	2c	NC_018589
**SLCC2482**	3	I	2b	NC_018591
**SLCC2540**	617	I	2b	NC_018586
**SLCC2755**	66	I	2b	NC_018587
**SLCC5850**	12	II	2a	NC_018592
**SLCC7179**	91	II	2a	NC_018593
**EGD-e**	12	II	2a	NC_022568

* MLST (Multilocus sequence typing-sequence type); NCBI (National Center for Biotechnology Information).

**Table 2 genes-10-01007-t002:** General features of *Listeria monocytogenes* strains.

Isolates Characteristics	*L. monocytogenes* Strain					
	E258	E313	E359	E362	E367	E916
Source	Biltong	Biltong	Biltong	Biltong	Biltong	Polony
Year isolated	2016	2015	2016	2015	2016	2015
Province (food establishment)	1 (butchery)	1 (butchery)	2 (retail outlet)	2 (retail outlet)	3 (retail outlet)	4 (retail outlet)
Genome length (bp) *	2,994,232	2,997,211	3,020,685	3,101,293	2,994,400	3,107,420
Number of contigs	38	48	127	44	42	39
G+C Content (%)	37.88	37.90	37.84	37.82	37.98	37.70
N50	411,134	411,134	63,798	480,372	437,780	521,621
No. of plasmids	0	0	0	0	0	0
Number of Proteins	2983	2985	2967	3070	2955	3110
*Listeria* pathogenicity islands (LIPI-1)	+	+	+	+	+	+
*Listeria* pathogenicity islands (LIPI-3)	+	+	+	-	-	+
Stress Survival Islet (SSI-1)	-	+	-	-	-	-
Stress Survival Islet (SSI-2)	-	-	-	+	+	-

**Table 3 genes-10-01007-t003:** PCR-serogroup and MLST.

Strain	Source	MLST	Lineage	PCR-Serogroup
**E258**	Biltong	1	I	IVb
**E313**	Biltong	1	I	IVb
**E359**	Biltong	876	I	IVb
**E362**	Biltong	121	II	IIa
**E367**	Biltong	204	II	IIa
**E916**	Polony	1	I	IVb

**Table 4 genes-10-01007-t004:** Scaffold length and predicted prophages of *Listeria monocytogenes* strains.

Strain	Scaffold Length (bp)	Prophage Number	Status	Size (Kb)	Number of Proteins	Position	Most Common Phage
**E258**	2,986,597	1	Questionable	10.7	17	39,802–50,530	PHAGE_Lister_A118_NC_003216(5)
		2	Intact	53.3	78	2,287,899–2,341,226	PHAGE_Lister_vB_LmoS_188_NC_028871(31)
		3	Incomplete	33	29	2,554,382–2,587,413	PHAGE_Lister_vB_LmoS_188_NC_028871(13)
		4	Intact	48.1	62	2,938,128–2,986,260	PHAGE_Lister_vB_LmoS_293_NC_028929(42)
**E313**	2,988,893	1	Intact	41.3	62	338–41,646	PHAGE_Lister_vB_LmoS_293_NC_028929(42)
		2	Incomplete	42.1	29	390,171–432,271	PHAGE_Lister_vB_LmoS_188_NC_028871(13)
		3	Intact	51.4	74	645,517–697,015	PHAGE_Lister_vB_LmoS_188_NC_028871(28)
		4	Questionable	10.7	17	2,935,515–2,946,243	PHAGE_Lister_A118_NC_003216(5)
**E359**	2,927,179	1	Intact	37.1	57	261,259–298,366	PHAGE_Lister_vB_LmoS_188_NC_028871(31)
		2	Intact	44.6	70	565,277–609,901	PHAGE_Lister_vB_LmoS_293_NC_028929(46)
		3	Questionable	10.7	17	2,778,084–2,788,812	PHAGE_Lister_A118_NC_003216(5)
**E362**	3,073,586	1	Incomplete	10.7	17	416,400–427,126	PHAGE_Lister_A118_NC_003216(6)
		2	Intact	32.2	45	1,028,426–1,060,665	PHAGE_Lister_A006_NC_009815(7)
		3	Intact	40.6	55	1,630,887–1,671,526	PHAGE_Lister_LP_101_NC_024387(40)
		4	Incomplete	26.2	18	2,743,121–2,769,384	PHAGE_Lister_A500_NC_009810(9)
		5	Incomplete	15.1	22	2,803,865–2,819,005	PHAGE_Lister_A118_NC_003216(10)
		6	Incomplete	27.4	39	3,010,996–3,038,428	PHAGE_Lister_A118_NC_003216(29)
**E367**	2,954,073	1	Questionable	10.7	17	40,491–51,217	PHAGE_Lister_A118_NC_003216(5)
		2	Questionable	42.8	64	568,244–611,055	PHAGE_Lister_A500_NC_009810(35)
		3	Incomplete	11.2	18	1,359,390–1,370,672	PHAGE_Psychr_pOW20_A_NC_020841(1)
**E916**	3,104,227	1	Questionable	10.7	17	99,555–110,283	PHAGE_Lister_A118_NC_003216(5)
		2	Intact	40.7	63	646,498–687,254	PHAGE_Lister_LP_030_2_NC_021539(50)
		3	Intact	41.4	58	822,319–863,785	PHAGE_Lister_vB_LmoS_188_NC_028871(46)
		4	Intact	48.8	76	1,932,379–1,981,196	PHAGE_Lister_B054_NC_009813(67)

Intact (score > 90 of the total number of CDS of the region); Questionable (score 70–90 of the total number of CDS of the region); Incomplete (score < 70 of the total number of CDS of the region).

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
