# Peer review of "Genomic Diversity of Common Sequence Types of Listeria monocytogenes Isolated from Ready-to-Eat Products of Animal Origin in South Africa"

_genes, 2019, doi:10.3390/genes10121007_

Round 1
Reviewer 1 Report
This is a well written and thorough study of 6 Listeria monocytogenes genomes from South Africa. Collecting this baseline data will be of great use for future studies and outbreaks within South Africa and also throughout the world.
The prevalence of resistance to the common food processing antiseptics seems significant and could potentially be highlighted in the abstract.
"core genome size of 1,753": Is this genes rather than kb? I expect genes?
"among our genomes might be influenced by presence of prophages as their length 
varied and did not occur in the same places in the genomes.": It should be easy to measure the % of phage sequence in each genome and see if the variation in genome size correlates with the % phage sequence by plotting % phage sequence versus genome size.
Author Response
Point 1:

Reviewer 2 Report
The manuscript “Whole genome sequence analysis of Listeria monocytogenes isolated from ready to eat meat products in South Africa” by Matle and others describes a study of the genomes of six isolate of L monocytogenes. Being able to identify and track potential new foodborne pathogens is a critical part of food safety management. Here the authors isolate, sequence, assemble, and describe six strains of Lmo from various locations in South Africa. In addition, the authors compare them to previously assembled Lmo genomes to ascertain among other things their pathogenic potential. It is a nice study that adds important information for both local and global awareness of Lmo and their impact on public health. Nevertheless, several concerns need to be addressed, predominantly for clarity to the readers. These concerns are listed below:
Major concerns:
+ The section 2.1 Sample Information is a rather short summary of what the authors have done. It would be essential for future work to report how, where (ideally with GPS coordinates), when precisely samples were collection, as well as how samples were collected, stored and transported prior to experimental procedures.
+ The entire results section is very concise. It would help the reader if the authors would clarify why certain experimental procedures were done and how this relates to the previous questions. What is the question the authors try to answer? Followed by a few final sentences per section describing how the question was answered. When discussing genes or gene groups, please clarify what these genes are, what their function is, what is known about them regarding pathogenic potential. How did non-pathogenic genes compare to other strains?
+ Figure 1 can be improved if a map of SA was shown, highlighting where the samples came from.
+ In the phylogenetic trees displayed somewhat confusing. There is inconsistent use of names of bacterial strains. The blue color of the strains from this study don’t really stand out from the black font of the other strains. Use of color to group different serotypes for instance would help guide the reader. There were no bootstrapping or other methods used to provide confidence that the tree shown is reliable. The trees are not compared to discuss whether they are the same, similar, or very different. This is needed to provide confidence in the data. In Figure 3, at a height level or 0.2, there are only three main clusters. How reliable are the diversification below 0.2?
+ Figure 4 is a little tough to comprehend. Is the GC content information relevant? Is there differing levels of % identity for the 6 strains? There appear to be several hypervariable regions or poorly assembled regions which overlap between the 6 strains. These are very interesting. What genes are here in other strains? Are these also poorly assembled or hard to assemble regions in other strains? Are these regions involved in the bacteria being able to adapt to antibiotic challenges? The genes which are highlighted are hard to read.
+ Why is the resolution of Figure 5 so low and what is exactly displayed in this figure?
+ It is not very clear how the conclusion as reported on lines 570-1, “Therefore, there is a need to determine the correct use of BC products in food processing facilities, as improper use can lead to resistance” was reached from the data presented in this manuscript.
Minor concerns:
+ Typo on line 84: “as powerful” should be “as a powerful”.
+ On line 101, “SA” is mentioned for the first time, without clarifying what it means.
+ Typo on line 441: “½a” should be “1/2a”
+ Type on line 575: “SS-2” should be “SSI-2”
Round 2
Reviewer 2 Report
The authors have addressed all concerns raised. I have no further concerns.